# Digital Workflow for Interim Prosthetic Rehabilitation Through the All-on-4 Concept Using 3D Printing Additive Process

**DOI:** 10.3390/jcm14238353

**Published:** 2025-11-24

**Authors:** Miguel de Araújo Nobre, Ricardo Almeida, Carlos Moura Guedes, Gonçalo Alvarez, Carolina Antunes, Ana Ferro, Mariana Nunes, Armando Lopes, João Rangel, João Pedro Martins, Diogo Santos, Miguel Gouveia

**Affiliations:** 1Research, Development and Education, MALO CLINIC, Avenida dos Combatentes, 43, 1600-042 Lisbon, Portugal; cantunes@maloclinics.com; 2Prosthodontics, MALO CLINIC, Avenida dos Combatentes, 43, 1600-042 Lisbon, Portugal; ralmeida@maloclinics.com (R.A.); cguedes@malolinics.com (C.M.G.); jrangel@maloclinics.com (J.R.); jmartins@maloclinics.com (J.P.M.); 3CERAMICS, CAD-CAM, MALO CLINIC, Avenida dos Combatentes, 43, 1600-042 Lisbon, Portugal; galvarez@maloclinics.com; 4Oral Surgery, MALO CLINIC, Avenida dos Combatentes, 43, 1600-042 Lisbon, Portugal; aferro@maloclinics.com (A.F.); mnunes@maloclinics.com (M.N.); alopes@maloclinics.com (A.L.); dsantos@maloclinics.com (D.S.); mgouveia@maloclinics.com (M.G.)

**Keywords:** All-on-4, implant-supported prosthesis, immediate loading, 3D printing, photogrammetry, prosthesis failure

## Abstract

**Background/Objectives**: Fully digital workflows used in implant dentistry have been evolving to improve rehabilitation times and patient satisfaction. More studies are necessary for full scientific validation. The aim of this study was to evaluate the 6-month outcome of implant-supported fixed prostheses for full edentulism rehabilitation using OnX Tough 2 resin (SprintRay) and the Pro 2 (SprintRay) 3D printer following the All-on-4 concept. **Methods**: This study included 16 patients (10 female, 6 male) with 20 implant-supported fixed full-arch prostheses following the All-on-4 concept (10 rehabilitations for each stackable guide and photogrammetry protocols). Primary The primary outcome measure was implant and prosthetic survival. The secondary outcome measures included manufacturing issues, prosthetic passive fit, marginal bone loss (MBL), mechanical and biological complications, modified plaque and bleeding indexes, pocket depths, patient subjective evaluation, and the Oral Health Impact Profile. **Results**: No patients were lost to follow-up. Two prostheses failed and 2 two implants were lost, resulting in a cumulative survival rate of 90% and 97.5% at 6 months for prostheses and implants, respectively. The mean MBL was 0.31 mm ± 0.52 mm at 4 months. The mechanical complications rate was 50% at patient level. One patient (6.3%) experienced one biological complication. The grades regarding “comfort of prostheses in the mouth” and “overall chewing feeling” were 9.35 ± 1.29 and 8.79 ± 1.67 out of 10, respectively. The mean total sum of the OHIP-14 was 1.61 out of 56. **Conclusions**: Implant-supported full-arch rehabilitations with fixed prostheses following the All-on-4 concept, through a fully digital workflow protocol, are a viable option in the short term.

## 1. Introduction

Complete edentulism is a public health problem, with an estimated average rate of 6.82% globally [1]. A potential viable treatment option for complete edentulism is represented by the All-on-4 concept (Nobel Biocare AB, Gothenburg, Sweden), validated for short-, middle- and long-term outcomes, with a cumulative survival rate of 93% in the mandible at 18 years, and 94.7% in the maxilla at 14 years [2,3]. The All-on-4 concept (Nobel Biocare AB) is based on the use of four implants as cornerstones (two posterior implants benefiting from distal tilting and two anterior implants axially inserted) in immediate function, for complete edentulism rehabilitation [2,3].

The last decade provided significant developments in digital tools for implant dentistry. A recent development consists of using fully digital workflows, using 3D imaging, guided implant insertion, intra-oral scanners for digital impressions, and computer-assisted design–computer-assisted manufacture (CAD-CAM) prosthesis [4,5,6]. The use of fully digital workflows provides an increased patient satisfaction (being less invasive) and decreased intervention times [7,8]. Fully digitally fabricated implant-supported crowns can be considered at least comparable to conventional and hybrid workflows in terms of time efficiency, production costs, precision, and patient satisfaction [9]. Nevertheless, more studies are necessary for full scientific validation.

In addition, there is currently a need for scientific evidence on interim fixed implant-supported prostheses produced through an additive process. A major advantage of additive processes is the customization options, allowing for an almost unlimited variety of printing shapes and complexity [10]. The comparison between additive and subtractive processes has been object of research in recent studies. On one hand, digital techniques provided superior quality, especially for long-term use, when compared to conventional methods [11]. On the other hand, the comparison between subtractive and additive manufacturing is still ongoing. Limited data on additive manufacturing implant-supported fixed dental prostheses makes the comparison to subtractive methods difficult [12,13]. An in vitro study comparing subtractive and additive CAD/CAM procedures for PMMA interim crowns concluded that printed interim crowns were more accurate than milled interim crowns, but despite larger internal dimensional variations for milled crowns, a non-significant difference was registered in fit [14]. For both subtractive and additive fabrication methods, post-processing may considerably increase the production time [12]. For the additive fabrication method, the accuracy of interim dental prostheses can be affected by a significant number of factors, including the material, prosthetic factor, the system (technique and printer), the parameters (layer thickness, printing orientation), postprocessing (rinsing, post-polymerization), and aging [15]. A recent evolution of the additive process is represented by the 3D Printing Workflow (SprintRay Inc., Los Angeles, CA, USA) for fixed hybrid dentures, using the OnX Tough 2 (SprintRay^®^) and the Pro 2 (SprintRay^®^) 3D printer. The OnX Tough 2 consists of a nanoceramic hybrid class II 3D printing resin, displaying an impact strength of 28 J/m, 7986 MPa flexural modulus, and 147 MPa flexural strength. The Pro 2 3D printer consists of a next-generation dental 3D printer that features a 35-micron optical panel technology and 385 nm UV-A light engine, enabling it to print 6+ full-arch prostheses in a single job or use the Arch Kit (SprintRay Inc., Los Angeles, CA, USA) to print up to two times faster. Nevertheless, there are currently no studies that validate the implementation of this protocol regarding full-arch rehabilitations.

The aim of this study was to evaluate the 6-month outcome of full-arch rehabilitations following the All-on-4 concept, using interim prostheses produced through a 3D printing additive process (Pro 2 3D printer, SprintRay^®^) and OnX Tough 2 resin (SprintRay^®^).

## 2. Methods

This study consisted of a prospective cohort clinical study, conducted between August 2024 and August 2025 in a private practice (Lisbon, Portugal). This study was approved by an independent ethical committee (Ethical Committee for Health, authorization No. 002/2024), and was written following the CARE guidelines (Appendix A) and all patients provided written informed consent. Patients in need of full-arch implant-supported rehabilitation were selected for intervention.

The study included 16 patients (10 females and 6 males), with a mean (standard deviation) age of 54.3 years (11.4 years), treated consecutively with 20 implant-supported prostheses *ad modum* the All-on-4 Concept (Nobel Biocare AB, Gothenburg, Sweden). A total of 10 rehabilitations (4 full-arch maxillary rehabilitations, 4 full-arch mandibular rehabilitations, and 1 double full-arch rehabilitation) were performed according to a stackable guide protocol, and 10 rehabilitations (4 full-arch maxillary rehabilitations, 2 full-arch mandibular rehabilitations, and 2 double full-arch rehabilitations) were performed using free-hand surgery and photogrammetry protocol. A total of 6 patients had systemic conditions, with 3 patients presenting more than one condition. These conditions consisted of cardiovascular conditions (n = 3), mental health conditions (n = 3), endocrine conditions (n = 2), osteoporosis (n = 1), and history of oncologic disease (n = 1). A total of 5 patients were smokers. Considering the opposing dentitions, 7 patients had natural teeth, 5 patients had implant-supported prostheses, 2 patients had a removable prosthesis, and 2 patients had a miscellaneous of implant-supported prostheses and natural teeth.

### 2.1. Inclusion and Exclusion Criteria

Patients were included in this study provided they presented enough bone quantity to be rehabilitated using standard dental implants (from 7 to 25 mm of length) and they provided written informed consent to participate in the study. Exclusion criteria included patients with the need for zygomatic implants to perform a full-arch rehabilitation, in active chemotherapy or radiotherapy, or unable to provide written informed consent.

### 2.2. Surgical and Prosthetic Protocols

The interventions were described in previous publications [2,3]. The interventions were performed through local anesthesia (articaine chlorydrate with epinephrine 1:100,000; Scandinibsa 2%, Inibsa Laboratory, Barcelona, Spain). Diazepam was administered to the patients before surgery (Valium 10 mg, Roche, Amadora, Portugal). Patients received antibiotic medications, corticosteroids, anti-inflammatories, analgesics, and antacid medications. The mucoperiosteal flap was raised along the ridge. Relieving incisions were made on the buccal aspect in the molar area for both protocols. For patients in the stackable guide surgical protocol [16], this allowed inserting and fitting the base guide to the buccal cortical. The 2.2 mm drills, together with guide fixation pins (BlueSkyBio, Libertyville, IL, USA), were used to fix the guide to the bone. The guide drove bone reduction and the remaining guides. Subsequently, a Piezomed SA-320 unit (W&H, Burmoos, Austria), together with a round handpiece bur (Axis Dental, Crissier, Switzerland) attached to a Rounger Beyer (Hu-Friedy, Chicago, IL, USA), was used to obtain a stable and flat bone surface. Following bone regularization, the implant guide was stacked over the base guide, allowing the guided placement of the implants in accordance with the planning in BlueSkyBio software (V4.9.4 64 bit, BlueSkyBio, Libertyville, IL, USA). For the photogrammetry protocol, the surgery was performed freehand. The implants in this study achieved over 35 N/cm insertion torque in both approaches. Following implant insertion, the abutments were connected, using multi-unit plus straight abutments with 2 mm height (Nobel Biocare AB) for the anterior axial implants and multi-unit plus 4 mm height with 30 degree angulation (Nobel Biocare, AB, Gothenburg, Sweden) for the posterior tilted implants. In the stackable guide protocol, the implant guide was removed after implant insertion, and the prosthetic guide was placed on top of the base guide, allowing relining of the pre-surgical 3D printed interim prosthesis (OnX Tough 2, SprintRay Inc., Los Angeles, CA, USA) in the previously planned position using a pink acrylic resin (Unifast, GC Corporation, Tokyo, Japan), followed by chairside finishing. The surgeon then sutured the flap through the use of non-resorbable sutures (3-0, B Braun Silkam, Aesculap Inc., Center Valley, PA, USA). For the photogrammetry protocol, following the suture, the scan bodies were placed, photogrammetry was performed, and the information was sent to the dental laboratory in order to print and characterize the interim prosthesis (OnX Tough 2, SprintRay). In both approaches, the interim prosthesis produced through a 3D printing additive process (Pro 2 3D printer, SprintRay^®^ Inc., Los Angeles, CA, USA) and using the OnX Tough 2 resin (SprintRay^®^) was connected on the day of surgery, achieving immediate function. Figure 1 and Figure 2 represent clinical cases performed with the stackable guide and photogrammetry surgical protocols, respectively, while Figure 3 details the different workflow for both groups.

### 2.3. Outcome Measures

The primary outcome measures were prosthetic and implant survival. Prosthetic survival was evaluated considering the necessity of replacing the prosthesis and implant survival was evaluated considering the implants’ function [2,3]: (a) implant fulfilled its purported function; (b) implant was stable when individually and manually tested; (c) absence of signs of persistent infection jeopardizing the implant outcome; (d) absence of radiolucent areas around the implants; (e) absence of aesthetic complaints from the prosthodontist and the patient; and (f) implant-supported fixed prosthesis providing good hygienic maintenance and comfort. The need to remove an implant was treated as a failure.

Secondary outcome measures included manufacturing issues, the passive fit of the prosthesis, the wear of the prosthesis, marginal bone loss, mechanical and biological complications, modified plaque and bleeding indexes, pocket depths, patient subjective evaluation, and the Oral Health Impact Profile (OHIP-14). Technical evaluation concerning manufacturing issues was registered by the dental laboratory team whenever the printing process of the prosthesis revealed any problems. The passive fit assessment was determined by unscrewing all prosthetic screws except one of the anterior abutments and performing periapical radiographs at 10 days post-surgery. The wear of the prostheses (maximal depth loss in mm) was assessed through the TRIOS Patient Monitoring (3shape, Copenhagen, Denmark). Intra-oral scans were performed at prosthetic placement and after 4–6 months. The intra-oral scans were compared for differences in matching. Pairs of scans were then superimposed, and wear was measured in millimeters. The highest wear value in each sextant was registered, and average values were estimated. For the assessment of marginal bone loss, periapical radiographs were taken at baseline and at the 4-month follow-up appointment, using the parallel technique with a film-holder (Super-Bite, Hawe Neos Dental, Bioggio, Switzerland), the position of which was adjusted manually for an orthogonal film position. Marginal bone loss (MBL) was determined by an outcome assessor through an image analysis software (iRYS version 14.0, MyRay; Imola, Italy). The reference point for the reading was the implant platform (the horizontal interface between the implant and the abutment). The clarity of implant threads was used to accept or reject the radiographs, given that this criterion enables both sharpness and the orthogonal direction of the radiographic beam towards the implant axis. The dimensions on the radiographs were calibrated using the distance between implant threads as a reference. Mechanical complications included loosening or fracture of prosthetic screws or abutments, and the fracture of the prosthesis, while biological complications comprised probing pocket depths >4 mm (applying a pressure of 0.25 N and measured at the vestibular, mesial, lingual, and distal aspects of the implant); abscess formation; fistulae formation; suppuration; and patient adverse soft tissue reaction. Plaque levels were evaluated through the modified plaque index (mPLI) [17] while bleeding indexes were evaluated through the modified bleeding index (mBI) [17]. Pocket depths were evaluated using a periodontal probe calibrated to 0.25 N and were measured at the vestibular, mesial, lingual, and distal aspects of the implant. The subjective evaluation was assessed through a visual analog scale (range: 0—poor; 10—excellent) concerning the patients’ comfort, and “overall chewing feeling” during daily food intake routines. The Oral Health Impact Profile, version 14 (OHIP-14) [18], estimated the impact of the rehabilitation on the patients’ quality of life during the follow-up (10 days and 2-, 4-, and 6 months), including functional limitations, physical pain, psychological, discomfort, physical disability, psychological disability, social disability, and handicap.

### 2.4. Statistical Analysis

The variables implant and prosthetic survival were analyzed, estimating cumulative survival rates through life tables (actuarial method). The average (95% confidence intervals) and standard deviations were calculated for the following: age, marginal bone loss (MBL), prosthetic wear, and patient evaluations. The mode was estimated for mPLI and mBI. Frequencies were estimated for manufacturing problems, biological complications, and mechanical complications. The statistical analysis in this study was performed using the SPSS for Windows version 26.0^®^.

## 3. Results

A total of 16 patients were rehabilitated with 20 full-arch prostheses supported by 80 implants. Ten (50%) rehabilitations followed a stackable guide surgical protocol, and another ten rehabilitations were performed according to a photogrammetry surgical protocol.

No patients were lost to follow-up. At the 6-month follow-up, two prostheses and two implants failed, rendering an overall cumulative survival rate of 90% (Table 1) and 97.5% (Table 2) for prostheses and implants, respectively.

No manufacturing issues were reported during the prostheses’ printing process, although one patient experienced an adhesion issue between the acrylic resin and the OnX Tough 2 resin, and one prosthesis and one stackable guide fractured during placement. These fractures were attributed to the thin width of both elements and were repaired in the laboratory. All prostheses were judged to be passively adjusted. Considering the evaluation of the prosthetic wear, a mean (95% confidence interval) prosthetic wear after 6 months of follow-up of 0.27 mm (0.1; 0.43) was obtained (Figure 4).

Of the 80 implants placed in this study, 74 had readable radiographs at 4 months. The mean (standard deviation) overall marginal bone loss at 4 months was 0.31 mm (0.52 mm) for implants and 0.31 mm (0.33 mm) at patient level (Table 3).

Twelve mechanical complications were registered in eight patients (50% rate at patient level): one prosthesis fracture in one patient at the 2-month follow-up leading to the first prosthetic failure (Figure 5a); one prosthesis fracture in one patient at the 2-month follow-up; one event of separation between the OnX Tough 2 resin and the acrylic resin in position FDI 12 (Figure 5b) and a crown fracture in position FDI 16 on the same prosthesis in one patient at the 4-month follow-up (both mended at the dental laboratory with further occlusion re-evaluation); one fracture of a maxillary prosthesis (leading to the second prosthetic failure—Figure 5c) and one fracture of a mandibular prosthesis (mended chairside) occurring in the same patient at 4 months (the patient admitted to not respecting the indications on the soft diet); five loose abutments (four patients, five implants) at the 2-month follow-up (solved by retightening the abutments and occlusion re-evaluation); and one loose abutment at the 6-month follow-up (solved by retightening the abutment and occlusion re-evaluation). The mean (standard deviation) follow-up time for the occurrence of mechanical complications and for the occurrence of prosthetic fractures was 3.1 months (1.4 months) and 3.6 months (0.8 months), respectively. Considering the groups, six patients had mechanical complications from the stackable guides protocol (n = 4 abutment screw loosening; n = 2 prosthetic fractures), while two patients in the photogrammetry protocol had mechanical complications (n = 1 abutment screw loosening and one prosthetic fracture in one patient; n = 2 prosthetic fractures in one patient).

One biological complication was experienced by one patient (6.3%; photogrammetry group), consisting of suppuration in one implant (1.3%) at the 10-day follow-up appointment. This complication was solved by antibiotic therapy. The overall modal value for mPLI and mBI was 1 (plaque only visible after running the probe through the mucosal margin and isolated bleeding spots visible, respectively), at the 10-day and 2-, 4-, and 6- month follow-up appointments (Figure 6 and Figure 7). No pocket depths > 4 mm were registered.

Regarding the patients’ subjective evaluation, the overall mean (standard deviation) grade regarding “comfort of prostheses in the mouth” and “overall chewing feeling” was 9.35 (1.29) and 8.79 (1.67) out of 10, respectively, at 6 months. These values registered a decrease from the 10-day evaluation to the 2-month evaluation, and then a general increase with the passing of time, with patients reporting values of 9.18 (1.08) and 8.25 (2.42) for “comfort of prostheses in the mouth” and “overall chewing feeling”, respectively, at 10 days; 8.16 (2.53) and 7.74 (2.34) at 2 months; and 8.68 (2.72) and 8.93 (1.87) at 4 months (Table 4).

The overall mean (standard deviation) total sum of the Oral Health Impact Profile, version 14 (OHIP-14) was 1.61 (0.20) at the 6-month follow-up. With the passing of time, patients referred to an increase in the quality of life, with mean values of 3.29, 3.03, 2.21, and 1.61 at 10 days and 2, 4, and 6 months, respectively (Table 5).

## 4. Discussion

The present study focuses on the 6-month follow-up of a new proposal for full-arch implant-supported fixed prosthetic rehabilitations using a fully digital workflow. Considering the outcomes of prosthetic survival, implant survival, and complications obtained in this study, this new proposal for full-arch implant-supported rehabilitation may be a viable treatment option, still requiring further validation in implant dentistry. Immediate loading protocols are currently used in implant dentistry, providing the advantages of an immediate restoration while not significantly deferring from delayed loading when comparing main outcome measures such as marginal bone loss [19].

A cumulative survival rate of 90% was obtained for prostheses at 6 months of follow-up. The comparison with other studies must be limited to the evaluation of the interim prosthesis, with prosthetic failures determined either by potential fractures that would imply prosthetic replacement or lack of prosthetic support due to implant failure, as assumed in the present study. Previous studies on full-arch rehabilitations in the mandible and maxilla using the same rehabilitation alternative registered an interim prosthesis fracture rate of 18.3% [2] and 13.3% [3], respectively, providing an interim prosthetic survival rate between 81.7% and 86.7%. Similarly, La Monaca et al. [20] registered an 88.2% cumulative interim prosthetic survival rate for full-arch rehabilitations, owing not to prosthetic fractures but to implant failures. In our study, prosthetic fractures and implant failures were the causes for prosthetic failures, a result that, considering similar conditions, is comparable to the previously cited studies [2,3,20].

The 97.5% cumulative implant survival rate at six months of follow-up registered in the present report is considered a good preliminary result. The result is within the range of previous investigations with similar rehabilitation procedures and follow-up, where a range from 96.7% to 100% cumulative survival rate was obtained at 6 months of follow-up [21,22,23], demonstrating that this protocol represents a stable and viable option for full-arch rehabilitations.

Regarding the wear of the prostheses, a mean value of 0.27 mm was obtained. In an in vitro study by Abdelfattah et al. [24], a mean (standard deviation) volumetric wear of 0.023 mm^3^ (0.007 mm^3^) was determined for the OnX resin. Myagmar et al. [25] registered in an in vitro study a range of wear volume loss between 0.10 and 0.44 mm (depending on the type of material), under a load of 49 N and after 60,000 cycles (equivalent to 3 months of loading [24]). Additionally, Stober et al. [26] reported a volumetric wear of the material that ranged between 8 and 114.6 mm^3^ × 10^−3^ (depending on the material), under a load of 40 N and after 100,000 cycles (corresponding to 5 months of loading [24]). The values obtained in these in vitro studies are lower than the one obtained in our study. However, these studies used finite element analysis to simulate real-life scenarios, with loads approximately six times weaker than the mean maximum bite force for adults (~295 N) [27]. Opposingly, the present study was performed in a real-life scenario, with prostheses being submitted to a strong subject effect and variety, according to different genders, with previously registered biting forces in the range between ~285 N to ~305 N [27].

A mean (standard deviation) MBL of 0.31 mm (0.52 mm) was obtained at the 4-month follow-up in the present study. This result is within the expected range for a full-arch rehabilitation, taking into consideration the 0.46 mm obtained at the 1-year follow-up for the same type of rehabilitation and dental implant model [21], and compares favorably to the MBL reported in other studies on full-arch implant-supported prostheses at the same follow-up period (0.61 to 0.66 mm [28,29]).

The mechanical complication rate was 50% at patient level. This percentage was within the expected range considering the inclusion of ~31% of patients with full-arch implant-supported prosthesis as opposing dentition. This setup was previously indicated as a risk indicator for the occurrence of mechanical complications [30] given the potential lack of proprioception by the patient and the absence of a shock-absorbing element from the materials, compared, for example, to natural teeth. A retrospective study by Shen et al. [31] assessed the clinical outcomes of implant-supported full-arch immediate prosthesis and observed a high prevalence of mechanical complications at 6 months of follow-up. Furthermore, a study by Chen et al. [32], where full-arch prefabricated and conventionally fabricated prosthesis were compared, reported a 12.8% rate at patient level. This result is lower when compared to our study. Nevertheless, this study [32] excluded patients with uncontrolled systemic conditions and heavy smokers. Several studies show that the presence of systemic conditions, such as neurologic conditions [33,34,35], anxiety [36], and the use of selective serotonin reuptake inhibitors for the treatment of depression [37], may be related to parafunctional habits, being well-documented that bruxism increases the risk of mechanical complications occurring [38,39]. The consistent high rate of mechanical complications presented in several studies [40] confirms the necessity of a controlled recall plan. In addition, it is important for patients to understand that the re-establishment of a functional dentition through immediate loading is not synonymous with ingesting a hard food diet without restrictions, considering the functional osseointegration period needed to be surpassed. Furthermore, additive manufacturing parameters such as the layer thickness, printing orientation, postprocessing (rinsing and post-polymerization), and aging, need to be controlled in order to decrease the probability of mechanical complications [15].

The low rate of biological complications (1.3% at implant level and 6.3% at patient level) is comparable with a previous study on the same type of rehabilitation and implant model [21] with 1-year of follow-up. Furthermore, Chen et al. [32] registered no biological complications at 6 months for full-arch rehabilitations. The low biological complication rate registered in our study highlights the importance of a maintenance protocol during the functional osseointegration period, considering our study included patients with systemic conditions [41,42,43,44,45,46,47,48,49] and smoking habits [50,51,52] that may increase the risk of biological complications. The remaining biological evaluation parameters (plaque and bleeding indexes) were also comparable to a previous study on the All-on-4 Concept [23]. Furthermore, Corbella et al. [53], in a study investigating the outcome of full-arch rehabilitations, evaluated plaque, bleeding and pocket depths: the distribution of plaque levels was characterized by a majority of patients (58%) with absence of plaque, followed by 29% of patients with abundance of soft matter (highest plaque accumulation), and 12.7% of patients with mild or visible plaque; on the other hand, the prevalence of mucositis was 4.9%, and bleeding occurred in 10% of patients [53]. Our study registered a different distribution, with a large majority of patients (93%) with mild or visible plaque accumulation, followed by 7% with no plaque accumulation, and no patients with the highest degree of plaque accumulation, while for bleeding, 21% of patients registered no bleeding, 57% registered isolated bleeding spots visible, and 21% registered bleeding forming a confluent line on the mucosal margin (no patients registered spontaneous bleeding). The different results may be explained by distinct backgrounds in oral health between Italian patients [53] and Portuguese patients. According to a dental health index study performed in 2020, Italy ranked first place among European countries, while Portugal was ranked fifth [54]. Nevertheless, it should be noted that the prevalence of these conditions may influence the occurrence of chronic conditions (such as peri-implant pathology) once the biological osseointegration is concluded, and therefore, strict maintenance, home-care, and recall measures should be taken for prevention.

The high patient satisfaction and very low impairment impact on patients’ quality of life represent promising results, comparing favorably with other studies on immediate function full-arch rehabilitations. De Araújo Nobre et al. [55] reported a 0.73 score in the OHIP-14 questionnaire after 6 months of treatment in a study with rehabilitations through the All-on-4 concept using PEEK prostheses. Legg et al. [56] reported an OHIP-14 score of 4.6 after treatment, representing a very low impairment impact on patients’ quality of life. In addition, Babbush et al. [57] reported that 75% of their sample rated their postsurgical discomfort as less than expected, and 70% reported less swelling than expected. In our study, a satisfaction rate of 99.2% (painful aching in the mouth) and 99.5% (uncomfortable eating any food) were reported regarding the “physical pain” parameters, which demonstrates that the application of the protocol used in the present study represents added value to the quality of life and satisfaction of the patients.

The limitations of this study include being a single-center study, the small sample size, the absence of sample size estimation and specific patient allocation to the two groups, the short follow-up, and only using descriptive analytical methods. Considering these limitations, the results should be cautiously interpreted. The small sample size could potentially impact the investigation due to the decrease in external validity. The short follow-up time, although in accordance with the main goal of investigating the outcome of interim prostheses during the implants’ functional osseointegration period (roughly the first 6 months post-treatment), may represent an overestimation of the outcomes. Considering the study objectives, no further plan exists to investigate the outcome of these rehabilitations, as these restorations are considered as an interim rather than definitive alternative. The strengths of the present study include the prospective design, the implementation of two different digital workflows, and the low rate of dropouts, rendering an increased internal validity. Future studies should focus on evaluating the outcomes of All-on-4 rehabilitations using a fully digital workflow protocol through an additive process in larger samples and reduced treatment times.

## 5. Conclusions

Within the limitations of the present study, it can be concluded that implant-supported full-arch fixed prostheses rehabilitations following the All-on-4 treatment concept through a fully digital workflow and using an additive process with 3D printed interim prostheses are a viable option in the short term. High prosthetic and implant survival rates, low marginal bone loss, low rates of biological complications, high patient satisfaction, and very low impairment impact on the patients’ quality of life support this conclusion. The workflows through stackable guides or photogrammetry were considered safe.

## Figures and Tables

**Figure 1 jcm-14-08353-f001:**
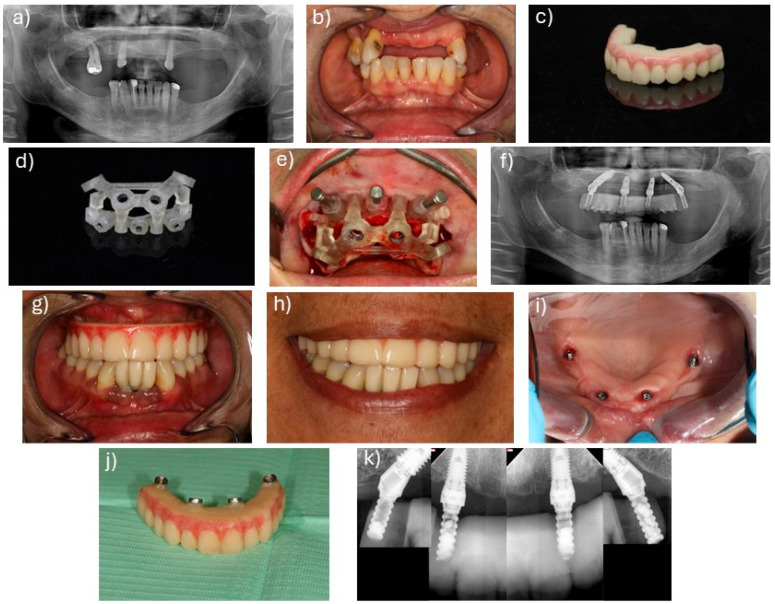
Clinical case representing a patient with a maxillary rehabilitation performed with the stackable guide surgical protocol. (**a**) Pre-treatment orthopantomography; (**b**) front view of the patient’s pre-treated oral cavity; (**c**) prosthesis printed with OnX Tough 2 resin with the Pro 2 3D printer; (**d**) printed stackable guides; (**e**) stackable guides placed in the maxilla; (**f**) post-treatment orthopantomography; (**g**) frontal view of the prosthesis in place; (**h**) patient’s smile after treatment; (**i**) implants’ occlusal view 6 months post-surgery; (**j**) prosthesis aspect after 6 months of placement; (**k**) 6-month apical radiographs of the placed implants.

**Figure 2 jcm-14-08353-f002:**
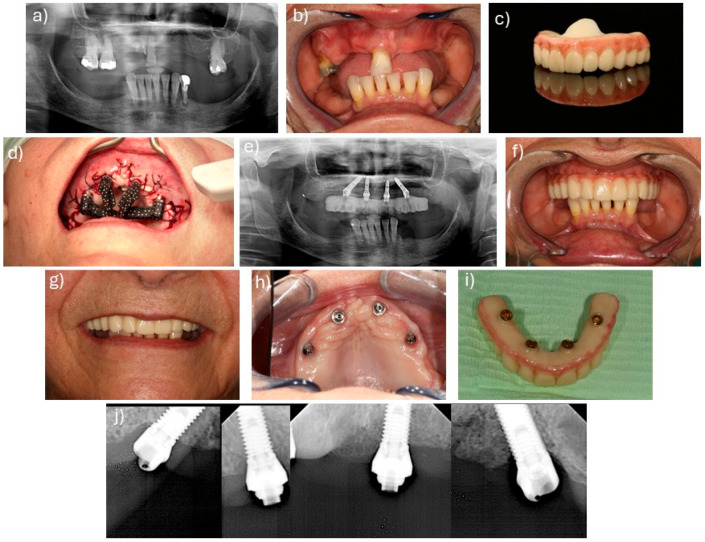
Clinical case representing a patient with a maxillary rehabilitation performed through a photogrammetry surgical protocol. (**a**) Pre-treatment orthopantomography; (**b**) frontal view of the patient’s pre-treated oral cavity; (**c**) prosthesis printed with OnX Tough 2 resin with the Pro 2 3D printer; (**d**) scan bodies placed for the photogrammetry surgical protocol; (**e**) post-treatment orthopantomography; (**f**) frontal view of the prosthesis in place; (**g**) patient’s smile after treatment; (**h**) implants’ occlusal view 6 months post-surgery; (**i**) Prosthesis aspect after 6 months of placement; (**j**) 6-month apical radiographs of the placed implants.

**Figure 3 jcm-14-08353-f003:**
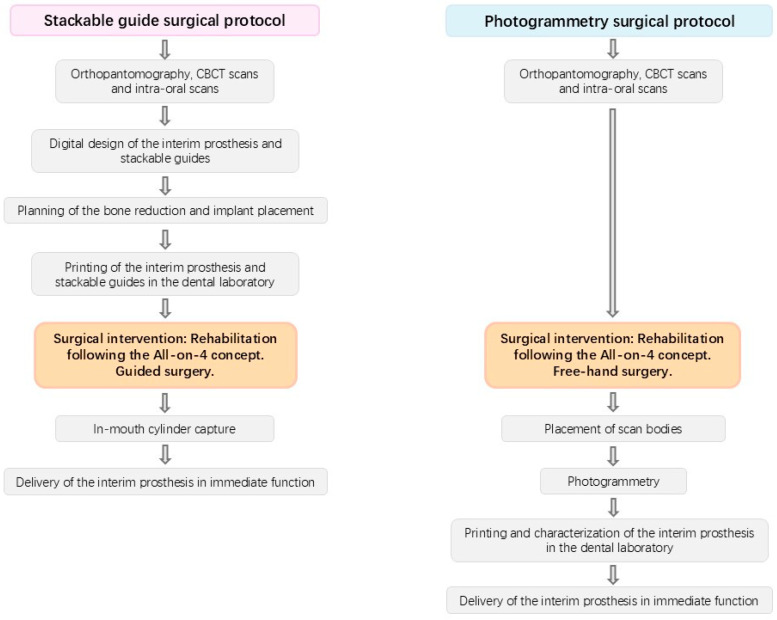
Flowchart illustrating the two rehabilitation workflows used in the present study: Stackable guides and photogrammetry.

**Figure 4 jcm-14-08353-f004:**
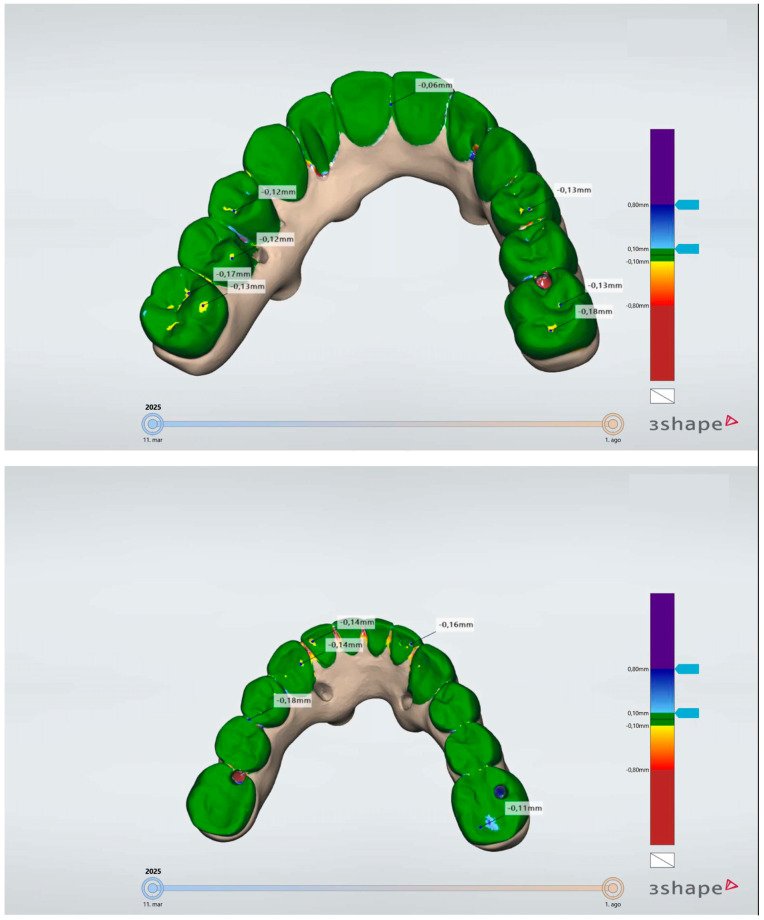
Wear measurements in a maxillary and mandibular prosthesis at the 6-month follow-up. Note the wear patterns measured in millimeters, with the highest value considered for each sextant.

**Figure 5 jcm-14-08353-f005:**
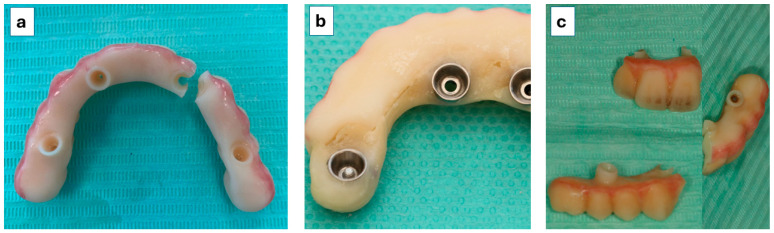
(**a**) Prosthetic failure at the 2-month follow-up; (**b**) prosthetic complication—separation between OnX Tough 2 and acrylic resins; (**c**) prosthetic failure at the 4-month follow-up.

**Figure 6 jcm-14-08353-f006:**
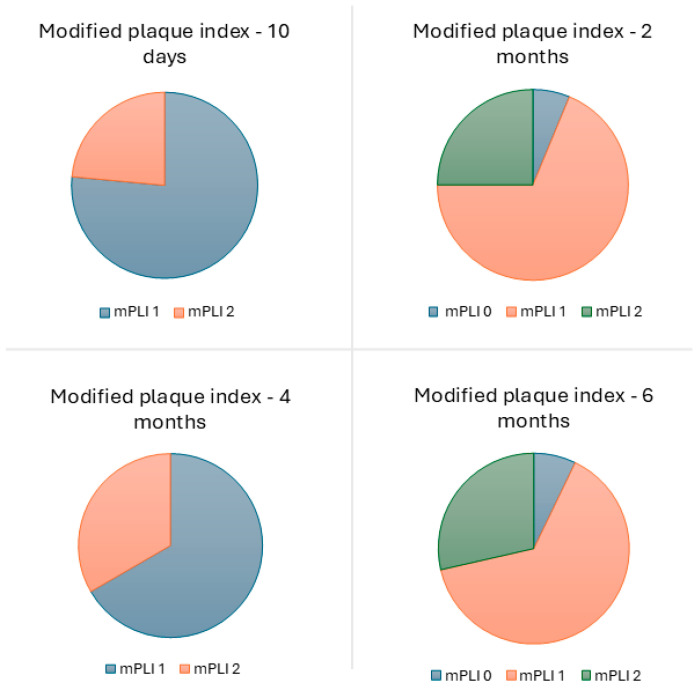
Modal value of the modified plaque index at the 10-day and 2-, 4-, and 6-month follow-ups. The modal value of the plaque index was the same for stackable guides and photogrammetry groups.

**Figure 7 jcm-14-08353-f007:**
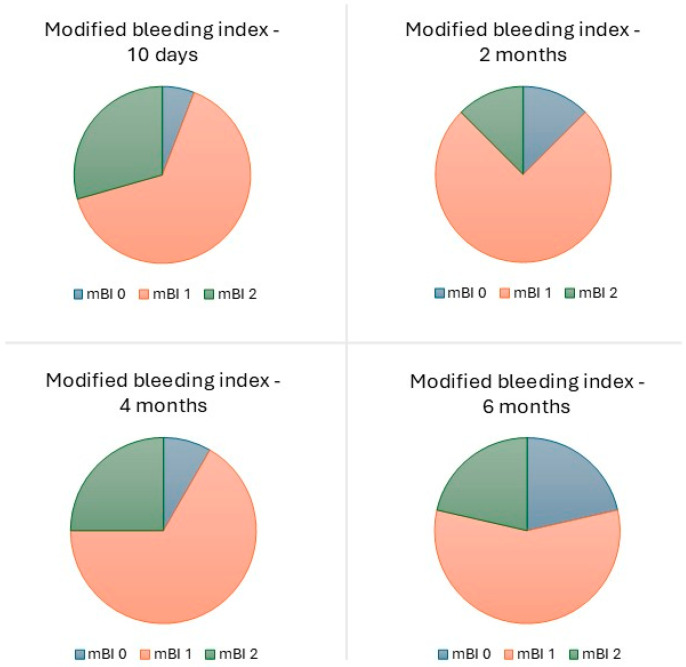
Modal value of the modified bleeding index at the 10-day and 2-, 4-, and 6-month follow-ups. The modal value of the bleeding index was the same for stackable guides and photogrammetry groups.

**Table 1 jcm-14-08353-t001:** Cumulative survival rate for prostheses inserted in this study, at the 6-month follow-up.

Time (Months)	Total Number of Prostheses	Lost to Follow-Up	Lost *	Survival Rate (%)	Cumulative Survival Rate (%)
0–2	20	0	1	95	95
2–4	19	0	1	95	90
4–6	18	0	0	100	90

* Both prosthetic failures were in the stackable guides group.

**Table 2 jcm-14-08353-t002:** Cumulative survival rate for implants placed in this study, at the 6-month follow-up.

Time (Months)	Total Number of Implants	Lost to Follow-Up	Lost *	Survival Rate (%)	Cumulative Survival Rate (%)
0–2	80	0	1	98.8	98.8
2–4	79	0	1	98.7	97.5
4–6	78	0	0	100	97.5

* One implant failure in each group.

**Table 3 jcm-14-08353-t003:** Marginal bone loss by implant and by patient, after 4 months of follow-up.

4-Month Follow-Up	Overall Bone Loss by Implant
Average (mm)	0.31 CI 95% [0.19–0.44] *
SD (mm)	0.52
Number (%) of implants with readable radiographs	74 (92.5%)
Frequencies	**N**
<0 mm	11
0 mm	22
0.1–1 mm	31
1.1–2 mm	10

* Stackable guides protocol: 0.26 mm (0.46). Photogrammetry protocol: 0.38 mm (0.39).

**Table 4 jcm-14-08353-t004:** Patients’ subjective evaluation for “comfort of prostheses in the mouth” and “overall chewing feeling” at the 10-day and 2-, 4-, and 6-month follow-ups (overall and distributed according to the workflow).

Patient Subjective Evaluation	10-Day Mean (Standard Deviation)	2-Month Mean (Standard Deviation)	4-Month Mean (Standard Deviation)	6-Month Mean (Standard Deviation)
*In-mouth comfort*				
Overall	*9.18 (1.1)*	8.16 (2.5)	8.68 (2.7)	9.35 (1.3)
Stackable guides	*9.37 (0.9)*	8.10 (3.0)	8.3 (3.2)	9.5 (1.4)
Photogrammetry	*9.10 (1.2)*	8.5 (1.6)	9.4 (0.8)	9.3 (1.0)
*Overall chewing feeling*				
Overall	8.25 (2.4)	7.74 (2.3)	8.93 (1.9)	8.79 (1.7)
Stackable guides	7.56 (2.8)	8.5 (2.5)	9.9 (0.3)	9.3 (1.3)
Photogrammetry	9.10 (1.6)	6.9 (1.7)	7.6 (2.3)	8.0 (1.7)

**Table 5 jcm-14-08353-t005:** OHIP-14 scores of the patients’ impact profile evaluation at the 10-day and 2-, 4-, and 6-month follow-ups (overall and distributed according to the workflow).

OHIP-14 Evaluation Parameters	10-Day Mean (Standard Deviation)	2-Month Mean (Standard Deviation)	4-Month Mean (Standard Deviation)	6-Month Mean (Standard Deviation)
*Functional limitation* Have you had trouble pronouncing any words? Have you felt that your sense of taste has worsened?	0.65 (1.11)	0.88 (1.11)	0.54 (1.08)	0.63 (1.08)
*Physical pain* Have you had painful aching in your mouth? Have you found it uncomfortable to eat any foods?	0.82 (1.12)	0.41 (0.74)	0.42 (0.81)	0.23 (0.56)
*Psychological discomfort* Have you been self-conscious? Have you felt tense?	0.44 (0.88)	0.69 (1.01)	0.50 (1.04)	0.24 (0.57)
*Physical disability* Has your diet been unsatisfactory? Have you had to interrupt meals?	0.76 (1.16)	0.56 (1.14)	0.42 (0.42)	0.33 (0.75)
*Psychological disability* Have you found it difficult to relax? Have you been a bit embarrassed?	0.21 (0.53)	0.38 (0.70)	0.25 (0.60)	0.07 (0.36)
*Social disability* Have you been a bit irritable with other people? Have you had difficulty doing your usual jobs?	0.35 (0.72)	0.09 (0.38)	0.08 (0.40)	0.10 (0.40)
*Handicap* Have you been unable to function? Have you felt life in general was less satisfying?	0.06 (0.24)	0.03 (0.17)	0.00 (0.00)	0.00 (0.00)
** *Total sum* **	3.29 (0.27) SG: 4.17 (0.34) P: 2.56 (0.22)	3.03 (0.28) SG: 2.67 (0.25) P: 3.56 (0.32)	2.21 (0.19) SG: 1.63 (0.18) 2.90 (0.35)	1.61 (0.20) SG: 1.01 (0.20) P: 2.50 (0.27)

SG: Stackable guides protocol; P: Photogrammetry protocol.

## Data Availability

Data is available at reasonable request to the authors.

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
