# Peer review of "Digital Workflow for Interim Prosthetic Rehabilitation Through the All-on-4 Concept Using 3D Printing Additive Process"

_jcm, 2025, doi:10.3390/jcm14238353_

Round 1
Reviewer 1 Report
Comments and Suggestions for Authors
Dear Authors,
I would like to thank you for submitting your manuscript and for your valuable contribution to the literature on digital workflows for full-arch implant-supported prosthetic rehabilitation. The topic is timely and relevant, particularly as additive manufacturing continues to expand its role in implant dentistry. The manuscript is well organized and the clinical data are clearly presented. However, several aspects could be refined to strengthen its scientific impact and clarity.
Below are my comments and suggestions for improvement.
Major Comments
-
Study Design and Sample Size
The study includes 16 patients and 20 prostheses, which represents a relatively small cohort. While acceptable for a preliminary investigation, please highlight this limitation more explicitly and discuss its impact on the generalizability of the findings. -
Follow-up Period
The 6-month follow-up period is relatively short for evaluating the survival and mechanical performance of full-arch prostheses. Please emphasize that these results represent short-term outcomes and clarify whether longer follow-up evaluations are planned. -
Comparison Between the Two Digital Workflows
Since two different protocols were implemented (stackable guide vs. photogrammetry), it would be valuable to include a more explicit comparison of clinical and prosthetic outcomes between these groups. Even descriptive trends would add depth to the discussion. -
Statistical Analysis
Please specify whether statistical tests were conducted to compare outcomes (e.g., complication rates, bone loss, satisfaction scores) between workflows. If statistical testing was not performed due to sample size, state this clearly as a limitation. -
Mechanical Complications
The mechanical complication rate (56.3%) is relatively high. Consider expanding the discussion to analyze possible contributing factors (e.g., occlusal scheme, opposing dentition, material thickness) and propose strategies to mitigate such issues in future work. -
Figures and Tables
-
Ensure that Figures 1 and 2 have adequate image resolution for publication.
-
Figure 3 (wear analysis) would benefit from clearer axis labeling and units.
-
Tables 1–3 could also include confidence intervals for clarity.
-
-
Ethical and Conflict of Interest Statements
Since the materials evaluated were donated by SprintRay, please clarify that the company had no role in study design, data collection, analysis, or manuscript preparation.
Minor Comments
-
Abstract
Consider briefly differentiating the two digital workflows in the abstract to provide context for the study design. -
Introduction
The introduction could be strengthened by discussing recent literature comparing additive and subtractive fabrication methods for interim prostheses and by addressing how printing accuracy and post-processing may affect clinical outcomes. -
Results
Provide standard deviations or confidence intervals consistently for all quantitative variables (e.g., comfort and chewing scores, marginal bone loss). -
Language and Style
The manuscript is generally well written, but a few minor typographical errors (e.g., “NAobel Biocare AB” on line 46) should be corrected. -
References
-
Ensure all references follow JCM citation formatting and that URLs and access dates are consistent.
-
It would also be useful to add a reference discussing loading protocols and their influence on bone remodeling and prosthetic performance. In particular, I recommend citing:
Crespi R, Fabris GBM, Crespi G, Toti P, Marconcini S, Covani U. Effects of different loading protocols on the bone remodeling volume of immediate maxillary single implants: A 2- to 3-year follow-up. Int J Oral Maxillofac Implants. 2019;34(4):953–962. doi:10.11607/jomi.6972.
This study provides valuable insights into how different loading times affect bone response and could enrich your discussion on immediate function and prosthetic loading in full-arch rehabilitations.
-
Author Response
Major Comments
- Study Design and Sample Size
The study includes 16 patients and 20 prostheses, which represents a relatively small cohort. While acceptable for a preliminary investigation, please highlight this limitation more explicitly and discuss its impact on the generalizability of the findings.
Response: The authors thank the Reviewer’s indication. The sample size was highlighted as a limitation in the Discussion section.
Changes: Lines 493-502
- Follow-up Period
The 6-month follow-up period is relatively short for evaluating the survival and mechanical performance of full-arch prostheses. Please emphasize that these results represent short-term outcomes and clarify whether longer follow-up evaluations are planned.
Response: The authors thank the Reviewer’s indication. The authors emphasized the short-term outcome. No longer follow-ups are planed as this represents an alternative for provisional restorations. This was also emphasized in the Discussion section.
Changes: Lines 493-502
- Comparison Between the Two Digital Workflows
Since two different protocols were implemented (stackable guide vs. photogrammetry), it would be valuable to include a more explicit comparison of clinical and prosthetic outcomes between these groups. Even descriptive trends would add depth to the discussion.
Response: The authors thank the Reviewer’s indication. The authors placed the results of both groups more explicitly as requested. The authors also added a flowchart to differentiate the workflow between both modalities. However, the authors did not perform any interpretation of these new segmented results, as the sample size is too small, no care was taken with the allocation of the patients between both groups (as it was not the objective of the study), and no inferential analysis was performed, therefore to discuss the trends without an inferential analysis would be a biased interpretation considering the limitations of the study design (all acknowledged in the Discussion section).
Changes: Lines 196 (new Figure 3- Flowchart), 284,286,315-324, 339-343, 350-351, 358-359, 362-363, 372-373 (new Table 4), 379-380, 493-495.
- Statistical Analysis
Please specify whether statistical tests were conducted to compare outcomes (e.g., complication rates, bone loss, satisfaction scores) between workflows. If statistical testing was not performed due to sample size, state this clearly as a limitation.
Response: The authors thank the Reviewer’s indication. The authors included the limitation exactly because of the reasons stated by the Reviewer.
Changes: 493-497
- Mechanical Complications
The mechanical complication rate (56.3%) is relatively high. Consider expanding the discussion to analyze possible contributing factors (e.g., occlusal scheme, opposing dentition, material thickness) and propose strategies to mitigate such issues in future work.
Response: The authors thank the Reviewer’s indication. First, the authors need to correct an error: effectively, there were mechanical complications in 8 patients and not nine as mentioned in the report, as 2 fractures occurred in the same patient. Despite the high mechanical complication rate (now at 50%), the majority were minor, representing loosening of prosthetic components. The authors attempted to present the potential contributing factors and strategies to mitigate such issues in future work in addition to the ones already stated.
Changes: Line 325, 446-452
- Figures and Tables
- Ensure that Figures 1 and 2 have adequate image resolution for publication.
- Figure 3 (wear analysis) would benefit from clearer axis labeling and units.
- Tables 1–3 could also include confidence intervals for clarity.
Response: The authors thank the Reviewer’s indications. Figures 1 and 2 were placed in the manuscript in low resolution so not to considerably increase the file size, please rest assured that the high resolution figures will be provided. Figure 3 clarity was improved by adding a comment on the legend as it is not possible to change the output of the file. Tables 1 and 2 cannot have confidence intervals because the method of estimation in life tables was the actuarial method and not Kaplan Meyer; Table 3 was improved with the inclusion of 95% confidence intervals.
Changes: Table 3, line 311
- Ethical and Conflict of Interest Statements
Since the materials evaluated were donated by SprintRay, please clarify that the company had no role in study design, data collection, analysis, or manuscript preparation.
Response: The authors thank the Reviewer’s indication. SprintRay had no role in study design, nor data collection, nor analysis, nor manuscript preparation. This was placed in the manuscript.
Changes: Lines 532-533
Minor Comments
- Abstract
Consider briefly differentiating the two digital workflows in the abstract to provide context for the study design.
Response: The authors thank the Reviewer’s indication. The Abstract was adapted as suggested.
Changes: Lines 22-23
- Introduction
The introduction could be strengthened by discussing recent literature comparing additive and subtractive fabrication methods for interim prostheses and by addressing how printing accuracy and post-processing may affect clinical outcomes.
Response: The authors thank the Reviewer’s indication. The Introduction was adapted as suggested comparing additive and subtractive fabrication methods for interim prostheses concerning printing accuracy and post-processing effects on clinical outcomes.
Changes: Lines 71-85
- Results
Provide standard deviations or confidence intervals consistently for all quantitative variables (e.g., comfort and chewing scores, marginal bone loss).
Response: The authors thank the Reviewer’s indication. Standard deviations and 95% confidence intervals for the average were introduced as requested.
Changes: Lines 315, 324,372,373,380
- Language and Style
The manuscript is generally well written, but a few minor typographical errors (e.g., “NAobel Biocare AB” on line 46) should be corrected.
Response: Thank you. Proof read and corrected.
Changes: Throughout the manuscript
- References
- Ensure all references follow JCM citation formatting and that URLs and access dates are consistent.
- It would also be useful to add a reference discussing loading protocols and their influence on bone remodeling and prosthetic performance. In particular, I recommend citing:
Crespi R, Fabris GBM, Crespi G, Toti P, Marconcini S, Covani U. Effects of different loading protocols on the bone remodeling volume of immediate maxillary single implants: A 2- to 3-year follow-up. Int J Oral Maxillofac Implants. 2019;34(4):953–962. doi:10.11607/jomi.6972.
This study provides valuable insights into how different loading times affect bone response and could enrich your discussion on immediate function and prosthetic loading in full-arch rehabilitations.
Response: The authors thank the Reviewer’s indication. References were proof read and confirmed to achieve format consistency and access dates as requested. The discussion on loading protocols and its influence in bone remodeling including the reference by Crespi et al. 2019 was added as requested.
Changes: Lines 386-389.
Reviewer 2 Report
Comments and Suggestions for Authors
Good morning Mr. Authors its a good job , but its necessary answer my commentaries.
Review commentaries
- Review the title recommended reduce
- Review the authors guidlines , the abstract have 313 words , the recomended is 250
- Review key words review MESH PUBMED
- In introduction the authors only name the Nobel Biocare , why use only Nobel Biocare ? give the funding for the research ? explane
- The Etic Commite its the University or Hospital ? explaene please.
- How calculate the sample ?
- How control the desertion?
- They used physical or digital survey
- Review the authors guidlines the tables have the gray contrast please review and put correct format
- Recommended added the CARE protocol .

Author Response
Review commentaries
- Review the title recommended reduce
Response: The authors thank the Reviewer’s indication. The title was reduced as requested.
Changes: Lines 2,3
- Review the authors guidlines , the abstract have 313 words , the recomended is 250
Response: The authors thank the Reviewer’s indication. The abstract was reduced to less than 250 words as requested.
Changes: Lines 16-37
- Review key words review MESH PUBMED
Response: The authors thank the Reviewer’s indication. MESH words were used as requested.
Changes: Lines 38-40
- In introduction the authors only name the Nobel Biocare , why use only Nobel Biocare ? give the funding for the research ? explain
Response: The authors thank the Reviewer’s queries. Nobel Biocare did not fund the study. The only reason we used Nobel Biocare implants is because we have been using them exclusively, by choice, for 30 years.
Changes: None.
- The Etic Commite its the University or Hospital ? explaene please.
Response: The authors thank the Reviewer’s query. It’s an Independent Ethical Committee recognized by the Portuguese National Order of Physicians and registered under the number 877.
Changes: None.
- How calculate the sample ?
Response: The authors thank the Reviewer’s query. The patients were consecutively included and no sample size calculations were performed, hence the authors did not perform any inferential analysis, only descriptive analysis. The authors introduced this as a limitation in the Discussion section.
Changes: Line 493-500
- How control the desertion?
Response: The authors thank the Reviewer’s query. The patients were contacted to participate in the study at the time of treatment planning and received a written informed consent explaining all the treatment and study phases. Then all clinical appointments were scheduled (surgery, prosthodontics, and maintenance) throughout the study period and all patients were contacted 48 hours prior to the clinical appointment in order to confirm the appointment. Nevertheless, this does not represent a different workflow from the one the authors regularly use with patients outside study protocols. It’s standard clinical practice.
Changes: None.
- They used physical or digital survey
Response: The authors thank the Reviewer’s query. The patients used a physical survey given by the treating Doctor.
Changes: None.
- Review the authors guidlines the tables have the gray contrast please review and put correct format
Response: The authors thank the Reviewer’s indication. The tables were adjusted as requested.
Changes: Lines 284,286,315,379,380
- Recommended added the CARE protocol .
Response: The authors thank the Reviewer’s indication. The CARE checklist was added as requested.
Changes: Supplementary file for review.
Round 2
Reviewer 2 Report
Comments and Suggestions for Authors
Good morning Mrs. Authors congratulations great job .